# Activation of AMPK/miR-181b Axis Alleviates Endothelial Dysfunction and Vascular Inflammation in Diabetic Mice

**DOI:** 10.3390/antiox11061137

**Published:** 2022-06-09

**Authors:** Chak-Kwong Cheng, Wenbin Shang, Jian Liu, Wai-San Cheang, Yu Wang, Li Xiang, Chi-Wai Lau, Jiang-Yun Luo, Chi-Fai Ng, Yu Huang, Li Wang

**Affiliations:** 1Department of Biomedical Sciences, City University of Hong Kong, Hong Kong 999077, China; superandy333@gmail.com; 2School of Biomedical Sciences, The Chinese University of Hong Kong, Hong Kong 999077, China; wenbinmedical@gmail.com (W.S.); wangyu.hmu@gmail.com (Y.W.); xiangli@hkbu.edu.hk (L.X.); chiwailau@cuhk.edu.hk (C.-W.L.); luojiangyun1985@gmail.com (J.-Y.L.); 3Department of Integration of Chinese and Western Medicine, School of Basic Medical Sciences, Peking University, Beijing 100871, China; liujianpku@bjmu.edu.cn; 4State Key Laboratory of Quality Research in Chinese Medicine, Institute of Chinese Medical Sciences, University of Macau, Zhuhai 519000, China; annacheang@um.edu.mo; 5Department of Surgery, The Chinese University of Hong Kong, Hong Kong 999077, China; ngcf@surgery.cuhk.edu.hk

**Keywords:** microRNA, diabetes mellitus, endothelial dysfunction, ROS, vascular inflammation, exercise, shear stress

## Abstract

Hyperglycemia in diabetes mellitus impairs endothelial function and disrupts microRNA (miRNA) profiles in vasculature, increasing the risk of diabetes-associated complications, including coronary artery disease, diabetic retinopathy, and diabetic nephropathy. miR-181b was previously reported to be an anti-inflammatory mediator in vasculature against atherosclerosis. The current study aimed to investigate whether miR-181b ameliorates diabetes-associated endothelial dysfunction, and to identify potential molecular mechanisms and upstream inducer of miR-181b. We found that miR-181b level was decreased in renal arteries of diabetic patients and in advanced glycation end products (AGEs)-treated renal arteries of non-diabetic patients. Transfection of miR-181b mimics improved endothelium-dependent vasodilation in aortas of high fat diet (HFD)/streptozotocin (STZ)-induced diabetic mice, accompanied by suppression of superoxide overproduction and vascular inflammation markers. AMPK activator-induced AMPK activation upregulated miR-181b level in human umbilical vein endothelial cells (HUVECs). Chronic exercise, potentially through increased blood flow, activated AMPK/miR-181b axis in aortas of diabetic mice. Exposure to laminar shear stress upregulated miR-181b expression in HUVECs. Overall, our findings highlight a critical role of AMPK/miR-181b axis and extend the benefits of chronic exercise in counteracting diabetes-associated endothelial dysfunction.

## 1. Introduction

The global prevalence and mortality rates of diabetes mellitus and its clinical complications, such as diabetic nephropathy and coronary artery disease, have been rising over decades [1]. Resulting from impaired insulin secretion and/or insulin resistance, type 2 diabetes mellitus is a progressive disease characterized by dysregulated metabolism of carbohydrates, lipids, and proteins [2]. Overt hyperglycemia, elevation of circulating blood glucose levels, causes burden on endothelial cells prompting the development of endothelial dysfunction during diabetes progression [3]. Endothelial dysfunction is characterized by dysregulated vascular tone, as reflected by imbalanced endothelium-dependent vasoconstriction and vasodilation, and disrupted redox balance [4]. Of note, endothelial dysfunction, oxidative stress, and inflammation are interrelated factors in the development of diabetes mellitus and cardiovascular diseases [5]. Increased production of reactive oxygen species (ROS) can aggravate vascular inflammation, and vice versa [6]. Enhanced oxidative and inflammatory burden drive endothelial dysfunction [7].

miRNAs are a class of short non-coding RNAs (~22 nucleotides), responsible for gene regulation in diverse cellular processes [8]. miRNA expression profiles in different tissues alter during the pathogenesis of various diseases. For example, the serum miRNA profiles significantly vary in diabetic and obese individuals [9]. Among miR-181 family members (miR-181a, miR-181b, miR-181c, and miR-181d), miR-181b was found negatively correlated with vascular inflammation and thrombogenicity in plasmas of type 2 diabetic patients [10]. Moreover, miR-181b has been shown to be an anti-inflammatory mediator in vasculature [11], particularly during atherogenesis [12]. However, whether miR-181b is aberrantly expressed in vascular cells during diabetes progression, and whether miR-181b upregulation ameliorates diabetes-associated endothelial dysfunction remains obscure. The detailed molecular mechanism underlying miR-181b-mediated vascular benefits also requires further exploration.

AMP-activated protein kinase (AMPK) is a key player in eukaryotes, responsible for the regulation of energy metabolism. More specifically, AMPK acts as a critical energy sensor which redirects metabolism from energy-consuming anabolism towards ATP-generating catabolism upon activation [13]. Additional to the effect on energy metabolism, AMPK activation (phosphorylation of threonine-172) is often considered as protective in the cardiovascular system. For example, metformin-mediated AMPK activation alleviates oxidative stress and endoplasmic reticulum (ER) stress in endothelial cells to restore vascular function [14]. Pharmacological activation of AMPK in vascular smooth muscle cells (VSMCs) restrains atherogenesis by upregulating autophagy [15]. Moreover, AMPK activation has been shown to inhibit cardiomyocyte hypertrophy in vitro and cardiac hypertrophy in vivo [16]. Previously, AMPK has been reported to be a downstream target of certain miRNAs (e.g., miR-148b and miR-451) in endothelial cells [17]. In addition, AMPK activation was shown to an upstream event of miRNA biogenesis in endothelial cells, where AMPK could phosphorylate nucleolin in nucleus to modulate miRNA expression [18]. It would be meaningful to study whether AMPK activation improves endothelial function by targeting miRNAs.

Vascular endothelial cells are constantly exposed to shear stress, the mechanical stimuli generated by blood flow [19]. Different shear patterns cause distinct responses in endothelial cells. Unidirectional laminar shear stress (LSS) in straight regions (e.g., thoracic aorta) of vasculature is anti-inflammatory, anti-atherogenic, and endothelium-protective [20]. Meanwhile, oscillatory shear stress (OSS) produced by disturbed flow in branched regions (e.g., aortic arch) of vasculature elicits pro-inflammatory and pro-atherogenic effects [21]. Certain miRNAs are considered mechanosensitive because their expressions in endothelial cells are altered upon exposure to different shear patterns. Therefore, shear stress-mediated alterations in miRNA expression are partially contributory to the regulation of endothelial homeostasis [22]. Additionally, exposure to shear stress modulated AMPK activation in endothelial cells [23].

In this study, we aimed to (i) investigate the level of miR-181b in arterial tissues of diabetic patients; (ii) study whether miR-181b upregulation is protective against diabetes-associated endothelial dysfunction; and (iii) identify underlying molecular mechanism and upstream inducer of miR-181b. Our findings might lead to new therapeutic strategy against diabetes-related vascular diseases.

## 2. Materials and Methods

### 2.1. Human Arteries

Human renal arteries (n = 12) were obtained from diabetic and non-diabetic patients subjected to nephrectomy. Table 1 reported the demographic characteristics of the patients. The study was conducted in compliance with the Declaration of Helsinki, and the protocol was approved by the ethics committee of clinic research, The Chinese University of Hong Kong (CUHK; approval number: 2014.468). These arterial samples were collected by Department of Surgery in Prince of Wales Hospital, CUHK. Some renal arteries from non-diabetic patients were freshly subjected to ex vivo treatment of AGEs (200 μg mL^−1^, 12 h; BioVision, Milpitas, CA, USA) and the AMPK activator 5-aminoimidazole-4-carboxamide riboside (AICAR; 2 mmol L^−1^, 12 h; Tocris, Milpitas, CA, USA) in Dulbecco’s modified Eagle’s medium (DMEM; 31885023; Gibco™, St. Louis, MO, USA), prior to RNA extraction and quantitative RT-PCR.

### 2.2. Animal Experimentation

All animal experiments were conducted in accordance with the Guide for the Care and Use of Laboratory Animals released by the National Institutes of Health, and the ARRIVE guidelines [24]. The animal study protocol was approved by the Animal Experimental Ethics Committee, CUHK (approval number: 16-226-CRF). Male C57BL/6 mice, db/db mice and db/m^+^ mice at 8–10 weeks old were supplied by the CUHK Laboratory Animal Service Center. Some C57BL/6 mice were fed a normal chow diet (Rodent diet with 10 kcal% fat; D12450B; Research Diets, Inc., New Brunswick, NJ, USA) or a HFD (Rodent diet with 60 kcal% fat; D12492; Research Diets, Inc.). After 6-week HFD feeding, the C57BL/6 mice were subjected to intraperitoneal (IP) injection of STZ (40 mg kg^−1^; Sigma-Aldrich, Burlington, VT, USA) for 5 consecutive days to induce type 2 diabetes mellitus [25]. The mice were sacrificed by CO_2_ anesthesia to dissect arteries for organ culture, functional assay by wire myography, ROS determination, RT-PCR and Western blotting. All mice were kept in well-ventilated caging systems at a constant temperature (23 ± 1 °C) and humidity (55 ± 5%) under a 12 h light/dark cycle. All mice were guaranteed free access to water and laboratory food pellets.

### 2.3. Blood Glucose Measurement

Glucose tolerance test (GTT) and insulin tolerance test (ITT) were conducted in mice post-fasting (8 h for GTT and 2 h for ITT, respectively). By oral administration of glucose (1.2 g kg^−1^) and/or IP injection of insulin (1 unit kg^−1^), the blood glucose levels were measured in venous blood from mouse tails at specified time points (0, 15, 30, 60, 90, and 120 min) [26].

### 2.4. Organ Culture of Arteries

After mice were sacrificed by CO_2_ suffocation, thoracic aortas were dissected out, and adhering connective tissues were carefully removed in ice-cold sterile PBS. Some aortas were cut into ring segments (~2 mm in length) for later functional assay by wire myography. The aortas were cultured in DMEM (31885023; Gibco™), supplemented with antibiotics and 10% FBS (16000044; Gibco™). To over-express miRNA mimics and inhibitors in freshly isolated aortas, miR-181b mimics (20 nmol L^−1^, QIAGEN, Hilden, Germany), miR-181b inhibitor (50 nmol L^−1^, 24 h; QIAGEN) or negative control (50 nmol L^−1^, QIAGEN) were transfected with Lipofectamine RNAiMax (Invitrogen, St. Louis, MO, USA) according to the manufacturer’s protocol [27]. After 24 h of transfection, the arteries were incubated with the AMPK activator AICAR (2 mmol L^−1^, 12 h; Tocris, Bristol, UK) in DMEM, prior to functional assay, ROS determination, and quantitative RT-PCR.

### 2.5. Functional Assay by Wire Myography

The ring segments of aortas (~2 mm) of mice were transferred to ice-cold and oxygenated Krebs solution containing (in mmol L^−1^): 119 NaCl, 4.7 KCl, 25 NaHCO_3_, 1.2 KH_2_PO_4_, 1 MgCl_2_, 2.5 CaCl_2_, and 11 D-glucose. The arterial segments were then individually mounted on the Multi Wire Myograph System (Danish Myo Technology, Hinnerup, Denmark) for measurement of isometric tension. The segments were initially stretched to an optimal baseline tension (aorta: 3 mN) and were allowed to equilibrate at 37 °C in Krebs solution under continuous oxygenation (95% O_2_, 5% CO_2_) for 1 h. The segments were precontracted by 60 mmol L^−1^ KCl and rinsed three times in Krebs solution. After that, phenylephrine (Phe; 3 μmol L^−1^) was added to precontract the segments, and endothelium-dependent relaxations (EDRs) were induced by the cumulative additions of acetylcholine (ACh; 3 nmol L^−1^ to 10 μmol L^−1^). Endothelium-independent relaxations were determined by cumulative addition of sodium nitroprusside (SNP; 1 nmol L^−1^ to 10 μmol L^−1^). The changes in isometric tension were monitored by PowerLab LabChart 7.0 system (AD Instruments, Dunedin, New Zealand) [28].

### 2.6. Lucigenin-Enhanced Chemiluminescence Assay

Superoxide anion production in mouse aortas were measured by lucigenin-enhanced chemiluminescence assay. Briefly, dissected mouse aortas were incubated in Krebs-HEPES solution containing (in mmol L^−1^): 99 NaCl, 4.7 KCl, 25 NaHCO_3_, 1 KH_2_PO_4_, 20 Na-HEPES, 1.2 MgSO_4_, 2.5 CaCl_2_ and 11 glucose, supplemented with diethyldithiocarbamic acid (1 mmol L^−1^; Sigma-Aldrich) and β-NADPH (0.1 mmol L^−1^; Sigma-Aldrich) at 37 °C for 45 min. The aorta were then transferred to vials containing lucigenin (10 μmol L^−1^; Sigma-Aldrich) in Krebs-HEPES solution [29]. Repeated measurements were recorded by the GloMax^®^ 20/20 Luminometer (Madison, WI, USA) in 1 min intervals for 10 min. The amount of superoxide anion produced was presented as real numbers in relative light units per mg of dry tissue [30].

### 2.7. ROS Determination by Dihydroethidium (DHE) Staining

After organ culture, some aortic rings were frozen in optimal cutting temperature compound (OCT; Sakura Finetek, Torrance, CA, USA) and later sliced to 10 μm sections by the CryoStar™ NX70 Cryostat (Thermo Fisher Scientific, Billings, MT, USA). ROS generation in the cross sections of mouse aortas were measured by DHE fluorescence staining, as previously described [31]. In brief, aortic cross sections were incubated with DHE (5 μmol L^−1^; Invitrogen) at 37 °C for 15 min in normal physiological saline solution (NPSS) containing (in mmol L^−1^): 140 NaCl, 5 KCl, 5 HEPES, 1 MgCl_2_, 1 CaCl_2_, and 10 glucose. Subsequently, the cross sections were rinsed thrice in NPSS. Fluorescent signals (DHE: excitation: 515 nm, emission: 585 nm; elastin autofluorescence: excitation: 488 nm, emission: 520–535 nm) were detected by the Olympus Fluoview FV1000 laser scanning confocal system (Olympus, Shinjuku City, Tokyo, Japan). DHE fluorescence signal was presented in real numbers.

### 2.8. Cell Culture

HUVECs (CC-2519; Lonza, Basel, Switzerland) were cultured in DMEM/F12 medium (11320033; Gibco™), supplemented with antibiotics, endothelial cell growth supplement (50 μg mL^−1^; Lonza) and 10% FBS (16000044; Gibco™) until confluency. HUVECs at passages four to seven were used for experiments. HUVECs were transfected with miR-181b mimics (20 nmol L^−1^), miR-181b inhibitor (50 nmol L^−1^) or negative control (50 nmol L^−1^) by using Lipofectamine RNAiMax (Invitrogen) following the manufacturer’s instructions. The cells were transfected and cultured for 24 h before pharmacological treatment. For pharmacological incubation, HUVECs were incubated with AGEs (200 μg mL^−1^, 12 h), the AMPK activator AICAR (2 mmol L^−1^, 4–12 h), and the AMPK inhibitor 9-beta-d-arabinofuranoside (ara-A; 1 mmol L^−1^, 12 h; Sigma-Aldrich).

### 2.9. Exercise Protocol

Male db/m^+^ and db/db mice were randomized into several groups and were allowed to train on a motorized running treadmill exercise system. For animal acclimatization, exercise intensity and duration were steadily increased during the first week of exercise training. The treadmill speed was initially set at 5 m min^−1^ for 30 min and later gradually increased to the target speed at 8 m min^−1^ for 30 min (240 m in total). The running exercise intensity was within the tolerance level of a mouse [32]. The mice were exercised daily, 6 days a week, for 7 consecutive weeks after 1 week of acclimatization training (8 weeks in total). Sedentary groups were kept on a non-moving treadmill for the same time duration as the exercised groups. Some diabetic mice were subjected to daily IP injection of the AMPK activator compound C (20 mg kg^−1^; Sigma-Aldrich) [33], for 4 consecutive weeks during the exercise protocol.

### 2.10. Hemodynamic Study

To generate LSS (12 dyn cm^−2^), the ibidi flow system (Gräfelfing), together with custom-built flow chambers were used. The flow system and computer control program were assembled according to the manufacturer’s instructions. HUVECs (5 × 10^5^ cells) were seeded on fibronectin-coated (50 μg mL^−1^) glass slides (75 mm × 38 mm; Corning, Corning, NY, USA). HUVECs were allowed to attach for 16 h. HUVECs were incubated in endothelial growth medium (EGM) supplemented with 2% FBS. The glass slides were mounted into the custom-built flow chambers and assembled to the ibidi flow system [34]. Some HUVECs were treated with ara-A (1 mmol L^−1^, 12 h). HUVECs were maintained at static condition, or exposed to LSS for 12 h, before protein and RNA extraction.

### 2.11. Quantitative RT-PCR

Total RNA from tissues and cells were extracted with TRIzol reagent (Invitrogen). For quantification of miRNA, cDNAs from miR-181b and U6 small nuclear RNA (snRU6) were synthesized by using the Taqman^TM^ miRNA Reverse Transcription kit (Applied Biosystems, Waltham, MA, USA). miR-181b expression was determined by TaqMan^TM^ miRNA assay kit (Applied Biosystems) in ABI ViiA7 system (Applied Biosystems) [35]. snRU6 was used as an internal control. Primer identification catalog numbers were: 001098 for hsa-miR-181b, 464957 for mmu-miR-181b, and 001973 for snRU6. For quantification of mRNA, cDNAs were synthesized by using the iScript^TM^ cDNA synthesis kit (Bio-Rad). Quantitative RT-PCR was performed by using SYBR Premix ExTaq (TaKaRa) in ABI ViiA7 system. GAPDH or Gapdh was used as an endogenous control. Human and mouse primer pairs used in the study were listed in Table 2 and Table 3, respectively.

### 2.12. Western Blotting

Tissues and cells were lysed in ice-cold 1X RIPA buffer (20-188; Millipore Corp.), supplemented with Phos-STOP phosphatase inhibitor (Roche, Basel, Switzerland) and Complete Protease Inhibitor cocktail (Sigma-Aldrich, St. Louis, MO, USA). Protein concentrations were measured by the BCA protein assay kit (Pierce Biotechnology, Waltham, MA, USA). Protein samples were mixed with loading buffer, supplemented with 5% β-mercaptoethanol, and were denatured at 95 °C for 5 min. Equal amount of protein samples were resolved by a 10% SDS-polyacrylamide gel and were transferred to an Immobilon-P polyvinylidene difluoride membrane (Millipore Corp., Burlington, VT, USA). The membrane was blocked by 3% BSA in TBS in the presence of Tween-20 (0.05%) for 30 min. The blots were subjected to overnight incubation at 4 °C with primary antibodies: anti-AMPKα (1:1000; 2532; Cell Signaling Technology, Danvers, MA, USA), anti-phospho-AMPKα at Thr172 (1:1000; 2535; Cell Signaling Technology), and anti-GAPDH (1:1000; 2118; Cell Signaling Technology). The membrane was washed and incubated with horseradish peroxidase-conjugated secondary antibodies (Cell Signaling Technology) at room temperature for 2 h. Protein bands were visualized by enhanced chemiluminescence reagent (Cell Signaling Technology), and were quantified by ChemiDoc^TM^ Imaging System (Bio-Rad, Hercules, CA, USA).

### 2.13. Experimental Blinding and Randomization

Experimental blinding was undertaken to reduce the risk of bias whenever possible in this study. The drugs used for the experiments were prepared by laboratory members who did not perform the experiments. Moreover, all animals were randomized before treatments.

### 2.14. Statistical Analysis

Data obtained from this study were presented in mean ± SD. Statistical analysis was performed by GraphPad Prism software (Version 8.0). Sample sizes were selected based on previous experiments. Statistical significance was determined by unpaired *t*-test and non-parametric Mann–Whitney test for comparison between two groups, whereas by Brown–Forsythe and Welch ANOVA followed by Games–Howell’s multiple comparisons for comparison among multiple groups. A *p* value < 0.05 indicates statistical significance.

## 3. Results

### 3.1. miR-181b Expression Reduced in Diabetic Condition

To evaluate the expression of miR-181b in the vasculature under diabetic condition, we first measured and compared miR-181b levels in renal arteries from non-diabetic and diabetic patients. miR-181b expression was significantly lower in the renal arteries of diabetic patients when compared to that of non-diabetic patients (Figure 1A). To confirm the presence of miR-181b in endothelial cells, we removed endothelium from the aortas of C57BL/6 mice [31], and measured miR-181b levels. Significant loss of miR-181b and endothelial cell-specific markers Cdh5 were observed in endothelium-removed mouse aortas (Appendix A). We next sought to identify which diabetes risk factor would modulate miR-181b expression. During diabetes, levels of AGEs elevate because proteins or lipids are glycated upon the exposure to sugars [36]. Accumulation of AGEs causes endothelial dysfunction and inflammation [37]. We, therefore, hypothesized that AGEs downregulate miR-181b in arteries and endothelial cells. As expected, incubation of AGEs (200 μg mL^−1^) suppressed miR-181b expression in the renal arteries from non-diabetic patients (Figure 1B), and in HUVECs (Figure 1C).

### 3.2. miR-181b Overexpression Improved Endothelial Function in Aortas of Diabetic Mice

To study the role of miR-181b in the vasculature under diabetic condition, we firstly induce type 2 diabetes mellitus in C57BL/6 mice by IP injection of STZ after HFD feeding. In the HFD/STZ-induced diabetic mice, glucose intolerance and insulin resistance were observed over 120 min (Appendix A). We dissected the aortas from normal chow-fed mice and diabetic mice for later functional assay in response to ACh, as an indication for endothelial function [38]. Compared to the aortas from normal chow-fed mice, the vascular function of aortas from HFD/STZ-induced diabetic mice were significantly impaired (Figure 2A,B). Transfection of miR-181b mimics (20 nmol L^−1^) or incubation of AICAR (2 mmol L^−1^) significantly reversed the impairment on EDRs in aortas from diabetic mice (Figure 2A,B). Transfection of miR-181b inhibitor (50 nmol L^−1^) ablated the vascular relaxing effects of both miR-181b mimics and AICAR (Figure 2A,B), suggesting that miR-181b might be downstream to AMPK activation in the vasculature. Meanwhile, endothelium-independent relaxation upon the cumulative addition of the nitric oxide (NO^•^) donor SNP was also measured to confirm that vascular smooth muscle function was unaltered [39]. Presence of miR-181b mimics, miR-181b inhibitor and AICAR did not alter SNP-induced endothelium-independent relaxation in aortas (Figure 2C,D).

### 3.3. miR-181b Overexpression Suppressed Vascular ROS in Aortas of Diabetic Mice

We next investigated how miR-181b overexpression improved endothelial function in the vasculature of diabetic mice. Since an imbalance between NO^•^ and ROS would aggravate endothelial dysfunction [40], we hypothesized that miR-181b might restrain ROS overproduction to improve endothelial function under diabetic condition. We therefore conducted lucigenin-enhanced chemiluminescence assay and DHE staining to determine ROS production in aortas of diabetic mice. The ROS production in aortas of HFD/STZ-induced diabetic mice were significantly higher in those of normal chow-fed mice (Figure 3A). Transfection of miR-181b mimics lowered ROS production in aortas of HFD/STZ-induced diabetic mice. Meanwhile, the presence of miR-181b inhibitor ablated the ROS-lowering effect of both miR-181b mimics and AICAR (Figure 3A). Similar findings were noted in the cross sections of mouse aortas, as shown by DHE staining (Figure 3B,C). These findings also suggested that miR-181b to might partially mediate the anti-oxidative effect of AMPK in vasculature under diabetic condition.

### 3.4. miR-181b Overexpression Inhibited Vascular and Endothelial Inflammation

We also hypothesized that miR-181b improved endothelial function through suppression of vascular inflammation, since multiple inflammatory mechanisms trigger endothelial activation and dysfunction to exacerbate progression of cardiovascular diseases [41]. We thereby investigated the expression of common inflammatory markers under diabetic condition in mouse aortas (i.e., Icam1, Vcam1, and Il-6) and in cultured human endothelial cells (i.e., ICAM1, VCAM1, and IL-6). The expression of inflammatory markers were upregulated in aortas of HFD/STZ-induced diabetic mice (Figure 4A–C), and in HUVECs treated with 200 μg mL^−1^ AGEs (Figure 4D–F). Transfection of miR-181b mimics downregulated the expression of inflammatory markers in both diabetic mouse aortas (Figure 4A–C), and AGEs-treated HUVECs (Figure 4D–F). Treatment with miR-181b inhibitor reversed the anti-inflammatory effect of miR-181b mimics and AICAR in the mouse aortas and HUVECs (Figure 4). These findings also implied that AMPK activation elicits anti-inflammatory effect in the vasculature and endothelial cells partially via miR-181b.

### 3.5. AMPK Activation Increased miR-181b Expression in Human Arteries and Endothelial Cells

We have shown that the presence of miR-181b inhibitor ablated the vascular relaxing, anti-oxidative and anti-inflammatory effects of the AMPK activator AICAR. To further verify our postulation that AMPK activation upregulates miR-181b expression in arteries and endothelial cells, we treated human renal arteries and HUVECs with AICAR. Presence of AICAR (2 mmol L^−1^) upregulated miR-181b levels in renal arteries from non-diabetic patients (Figure 5A), and in HUVECs (Figure 5B). Moreover, presence of the AMPK inhibitor ara-A (1 mmol L^−1^) reversed the upregulating effect on miR-181b of AICAR in HUVECs (Figure 5C). These results indicated the AMPK/miR-181b axis in endothelial cells.

### 3.6. Chronic Exercise Activated AMPK/miR-181b Axis in Diabetic Mice

We next sought to identify potential means that could activate the AMPK/miR-181b axis in the vasculature under diabetic condition. Exercise is generally considered as beneficial to the cardiovascular health of diabetic patients [42], and chronic exercise has been previously shown to activate AMPK in the arteries of diabetic mice [32]. We, therefore, postulated that chronic exercise could increase miR-181b expression in arteries via AMPK activation. Hence, we subjected non-diabetic db/m^+^ mice and diabetic db/db mice to chronic exercise on a motorized running treadmill (Figure 6A). We used another widely used mouse model of type 2 diabetes mellitus to confirm the presence of AMPK/miR-181b axis upon exercise [43]. Chronic exercise induced AMPK activation and miR-181b upregulation in the aortas of db/m^+^ and db/db mice, when compared to the sedentary groups (Figure 6B–D). IP injection of the AMPK inhibitor compound C (20 mg kg^−1^) ablated AMPK activation and miR-181b upregulation in the aortas of exercise-trained db/db mice (Figure 6B–D). These findings suggested that chronic exercise upregulates miR-181b through AMPK activation.

### 3.7. LSS Activated AMPK/miR-181b Axis in Endothelial Cells

The benefits of exercise are mediated by cascades of physiological, molecular and cellular processes in the body [44]. During exercise, the systemic blood flow increases, leading to a higher shear stress on endothelial cells of the vasculature [45]. Therefore, we postulated that the increased shear stress would partially contribute to the activated AMPK/miR-181b axis in endothelial cells during exercise. To verify, we conducted in vitro hemodynamic study to mimic the increased shear stress (i.e., LSS) on human endothelial cells (Figure 7A). Exposure to LSS for 12 h induced AMPK activation and miR-181b upregulation in HUVECs (Figure 7B–D). Meanwhile, the presence of the AMPK inhibitor ara-A (1 mmol L^−1^) restrained both AMPK activation and miR-181b upregulation in HUVECs (Figure 7B–D), implying an LSS/AMPK/miR-181b signal transduction axis in endothelial cells.

## 4. Discussion

Among miR-181 family members, miR-181b has been shown to modulate endothelial activation and function to suppress vascular inflammation. Meanwhile, other members were mainly found in non-endothelial cells (e.g., VSMCs and immune cells) to regulate vascular inflammation [46]. Our current study reported that miR-181b level decreased in the renal arteries of diabetic patients. Our results suggested that miR-181b attenuated endothelial dysfunction through suppressing ROS generation and vascular inflammation in arteries of diabetic mice, hinting that miR-181b upregulation is protective against diabetes-associated vascular dysfunction. AMPK activation was shown to be an upstream event to miR-181b upregulation, where chronic exercise could, possibly via increased blood flow, activate the AMPK/miR-181b axis in endothelial cells (Graphical Abstract). Our findings also link mechano-stimulation to miRNA modulation in endothelial cells.

Under diabetic conditions, elevated AGEs are associated with insulin resistance [47]. AGEs result from the nonenzymatic glycation and oxidation of proteins and lipids [36]. AGEs can accumulate in the vascular wall, where they can perturb endothelial cell surface and injure endothelial cells by causing oxidative stress and inflammation [37]. Coherent to the finding in the renal arteries of diabetic patients, incubation of AGEs caused miR-181b downregulation in renal arteries of non-diabetic patients and in cultured endothelial cells. However, we cannot rule out the possibility that other factors under diabetic conditions might also suppress miR-181b expression in endothelial cells. Previously, AGEs have been shown to suppress AMPK activation in HUVECs [48]. Therefore, it is reasonable to postulate that AGEs might downregulate miR-181b expression in endothelial cells via an AMPK-dependent manner.

We provided experimental evidence that transfection of miR-181b mimics improved endothelial function, inhibited vascular ROS production, and restrained vascular inflammation in aortas of HFD/STZ-induced diabetic mice. Coherently, miR-181b overexpression limited the expression of inflammatory markers in AGEs-treated HUVECs. Diabetes mellitus is characterized by chronic vascular inflammation and, hence, a higher risk of diabetic vasculopathy and atherosclerosis [10]. Our results, therefore, provided therapeutic insights that miR-181b upregulation might be beneficial against diabetic vascular diseases. Hyperglycemic conditions impair endothelial function by causing enhanced oxidative stress and inflammation [49], where oxidative stress and inflammation are often interrelated factors aggravating disease progression [6]. A lower oxidative stress limits the quenching of NO^•^, where higher bioavailability of NO^•^ accounts for the improved endothelial function [50]. Meanwhile, presence of miR-181b inhibitor inhibited the vascular relaxing, anti-oxidative and anti-inflammatory effects of the AMPK activator AICAR, implying that miR-181b might be downstream to AMPK activation. Previously, circulating miR-181b was shown to suppress vascular inflammation via PTEN and KPNA4 [10]. However, whether miR-181b upregulation in endothelial cells ameliorates vascular inflammation under diabetic condition through similar mechanisms requires further study.

In addition, our findings suggest that AICAR-induced AMPK activation upregulated miR-181b expression in human renal arteries and endothelial cells. AMPK activation was confirmed to elicit anti-oxidative and anti-inflammatory effects in the vascular system [51]. Our findings extended the molecular mechanism underlying AMPK activation, where activation of the AMPK/miR-181b axis could elicit beneficial effect on endothelial function under hyperglycemic conditions. Previously, AMPK was shown to be upstream to the biogenesis and maturation of certain miRNAs (e.g., miR-93 and miR-484), through phosphorylation of nucleolin in nucleus, in endothelial cells [16]. Future study shall investigate whether AMPK activation upregulates miR-181b expression in endothelial cells through similar or different mechanisms.

To provide more therapeutic insights into the activation of AMPK/miR-181b axis, we questioned whether certain physiological means could activate the axis under diabetic conditions. Our previous study showed that chronic exercise could induce AMPK activation in the aortas of diabetic mice [32]. Consistently, our in vivo data showed that chronic exercise promoted AMPK activation and miR-181b upregulation in mouse aortas. Injection of the AMPK inhibitor compound C reduced miR-181b upregulation in the aortas of diabetic mice, indicating AMPK activation is an upstream event to miR-181b upregulation during exercise. Notably, our present study extends the molecular mechanism and vascular benefits of physical exercise. Moreover, our findings also imply the therapeutic potential and importance of physical activity in counteracting diabetes-associated vascular complications.

However, physical exercise induces a wide spectrum of downstream responses in our body [52]. During exercise, both cardiac output and systemic blood flow increase in our body [53]. Particularly, the increased blood flow would exert a higher LSS on the exposing endothelial cells in the vasculature [45]. Additionally, LSS promotes AMPK activation in endothelial cells [54]. Our data, therefore, show an additional molecular mechanism of LSS-induced AMPK activation, and suggest a beneficial exercise/LSS/AMPK/miR-181b mechanotransduction cascade in endothelial cells (Graphical Abstract). However, we cannot rule out the contribution of other proteins and signaling pathways. Future extensive efforts are still needed to explore the comprehensive mechanism underlying LSS-induced miR-181b upregulation. Certain miRNAs have been considered mechanosensitive to mediate inflammation, proliferation and permeability of endothelial cells [22], where miR-181b was previously not regarded as a mechanosensitive miRNA [55]. Our results extend the profile of mechanosensitive miRNAs and the role of miR-181b against endothelial dysfunction.

## 5. Conclusions

In summary, we showed that miR-181b level decreases in arteries of diabetic patients. miR-181b alleviates endothelial dysfunction, vascular inflammation, and oxidative stress in diabetic mice. AMPK activation upregulates miR-181b level in endothelial cells. In addition, chronic exercise, potentially through increased blood flow, activates AMPK/miR-181b axis in endothelial cells. These findings suggest a beneficial role of AMPK/miR-181b axis against diabetes-associated endothelial dysfunction.

## Figures and Tables

**Figure 1 antioxidants-11-01137-f001:**
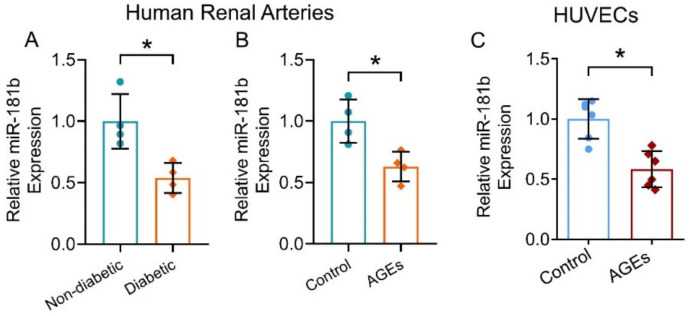
miR-181b expression in human renal arteries and endothelial cells. RT-PCR on (**A**) miR-181b level in renal arteries from non-diabetic and diabetic patients, and on (**B**) miR-181b level in renal arteries from non-diabetic patients treated with 200 μg mL^−1^ AGEs for 12 h. n = 4 per group. (**C**) RT-PCR on miR-181b expression in HUVECs treated with 200 μg mL^−1^ AGEs for 12 h. n = 6 per group. Data are mean ± SD. * *p* < 0.05 (unpaired *t*-test and nonparametric Mann–Whitney test).

**Figure 2 antioxidants-11-01137-f002:**
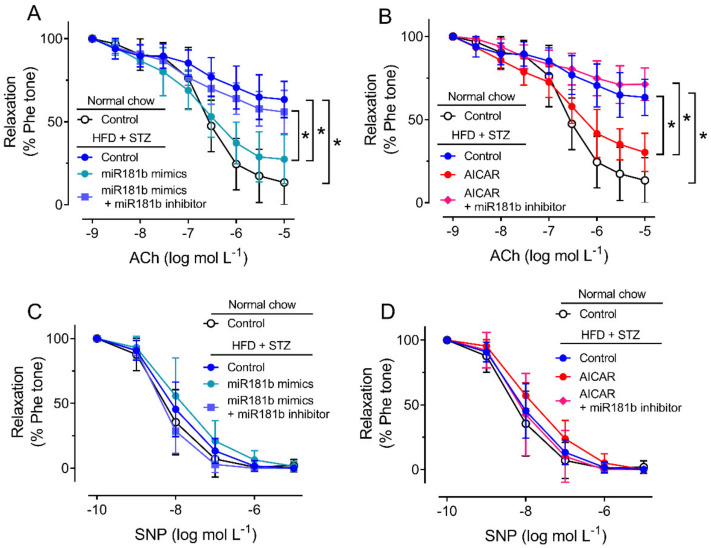
Effects of miR-181b mimics and inhibitor on vascular function. (**A**,**B**) Wire myography on EDRs of aortic rings from normal chow-fed mice and HFD/STZ-induced diabetic mice, upon (**A**) transfection of miR-181b mimics or (**B**) treatment with AICAR. (**C**,**D**) Wire myography on SNP-induced endothelium-independent relaxations of aortic rings from normal chow-fed mice and HFD/STZ-induced diabetic mice, upon (**C**) transfection of miR-181b mimics or (**D**) treatment with AICAR. n = 6 per group. Data are mean ± SD. * *p* < 0.05 (Brown-Forsythe and Welch ANOVA and Games–Howell’s multiple comparisons).

**Figure 3 antioxidants-11-01137-f003:**
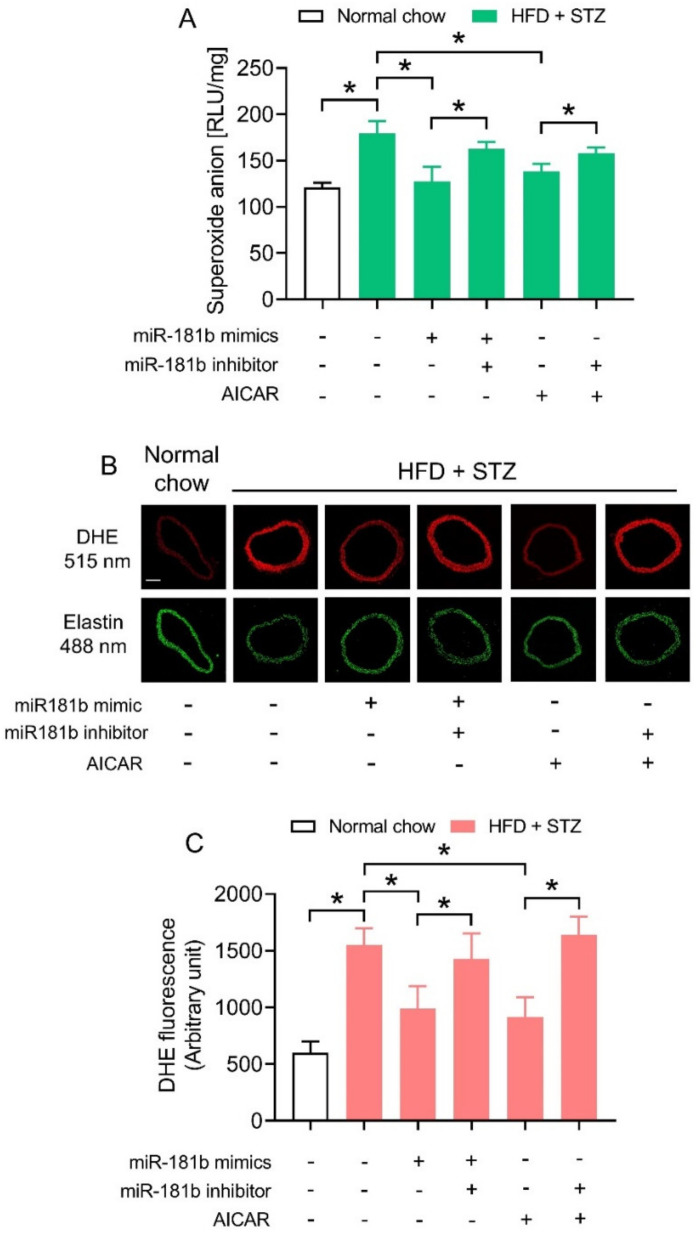
Effects of miR-181b mimics and inhibitor on vascular ROS production. (**A**) Lucigenin-enhanced chemiluminescence assay on the ROS levels in aortas of normal chow-fed and diabetic mice. (**B**) Representative images of DHE staining on ROS production in cross sections of mouse aortas. Scale bar = 200 μm. (**C**) Quantification of DHE fluorescence. n = 6 per group. Data are mean ± SD. * *p* < 0.05 (Brown–Forsythe and Welch ANOVA and Games-Howell’s multiple comparisons).

**Figure 4 antioxidants-11-01137-f004:**
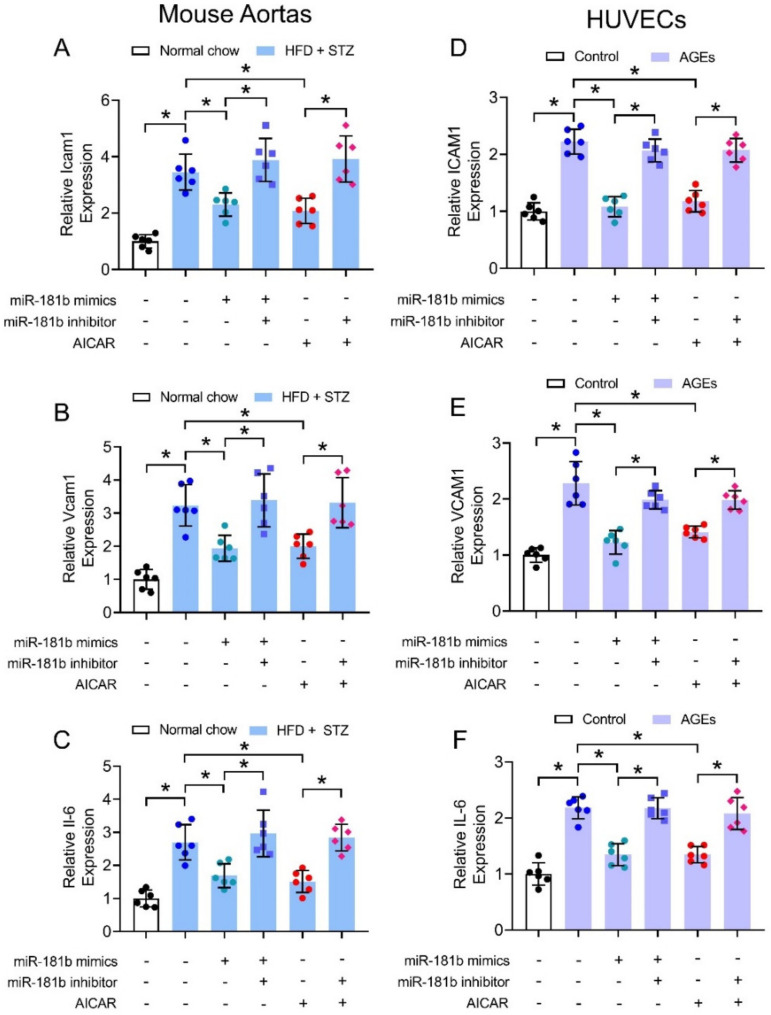
Effects of miR-181b mimics and inhibitor on vascular and endothelial inflammation. RT-PCR on expression of inflammatory markers (**A**) Icam1, (**B**) Vcam1, and (**C**) Il-6 in aortas of normal chow-fed and diabetic mice. RT-PCR on expression of inflammatory markers (**D**) ICAM1, (**E**) VCAM1, and (**F**) IL-6 in HUVECs treated with 200 μg mL^−1^ AGEs for 12 h. n = 6 per group. Data are mean ± SD. * *p* < 0.05 (Brown-Forsythe and Welch ANOVA and Games-Howell’s multiple comparisons).

**Figure 5 antioxidants-11-01137-f005:**
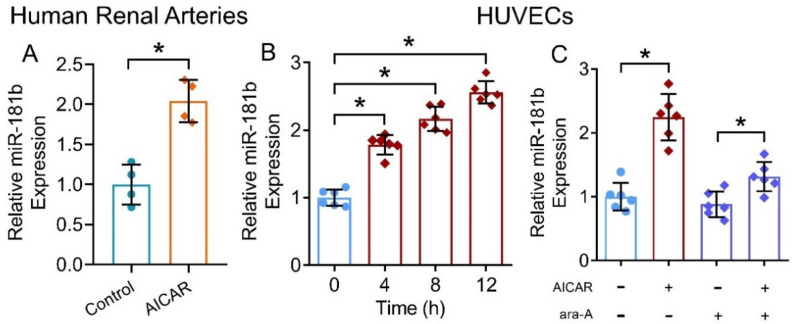
Effects of AMPK activation on miR-181b level in human arteries and endothelial cells. (**A**) RT-PCR on miR-181b expression in AICAR-treated renal arteries from non-diabetic patients. n = 4 per group. (**B**) RT-PCR on the time-dependent effect of AICAR on miR-181b expression in HUVECs. n = 6 per group. Data are mean ± SD. * *p* < 0.05 (unpaired *t*-test and nonparametric Mann–Whitney test). (**C**) RT-PCR on miR-181b level in HUVECs treated with 1 mmol L^−1^ ara-A for 12 h. n = 6 per group. Data are mean ± SD. * *p* < 0.05 (Brown–Forsythe and Welch ANOVA and Games–Howell’s multiple comparisons).

**Figure 6 antioxidants-11-01137-f006:**
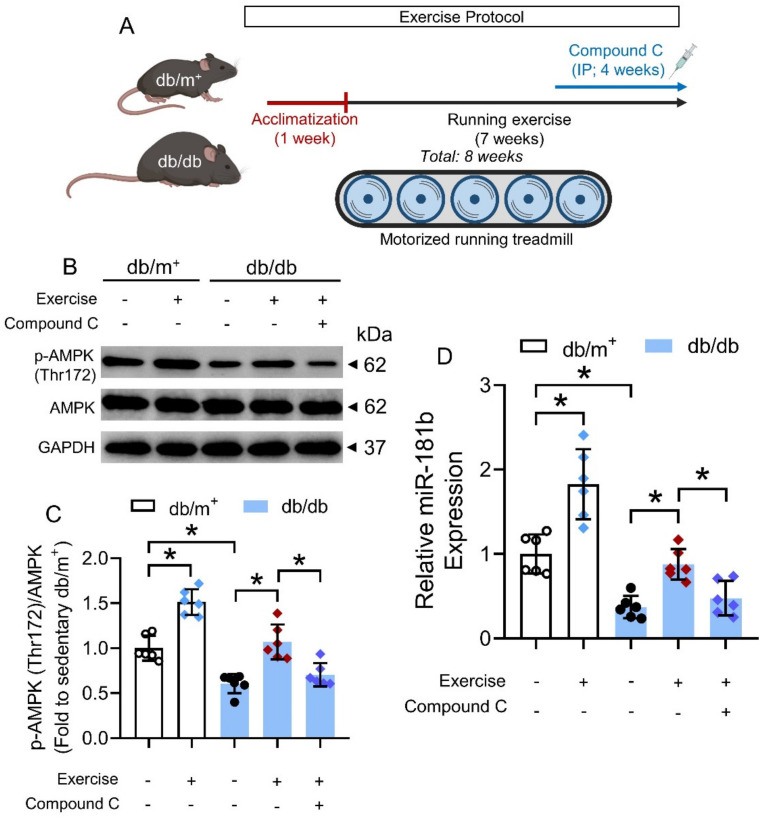
Effects of chronic exercise on AMPK/miR-181b axis in aortas of diabetic mice. (**A**) Schematic overview of chronic exercise training in diabetic mice. (**B**) Representative Western blots, and (**C**) quantification of Western blotting on expression of AMPK and p-AMPK at Thr172 in aortas of sedentary or exercise-trained diabetic mice. n = 6 per group. (**D**) RT-PCR on miR-181b expression in aortas of sedentary or exercise-trained diabetic mice. n = 6 per group. Data are mean ± SD. * *p* < 0.05 (Brown-Forsythe and Welch ANOVA and Games–Howell’s multiple comparisons).

**Figure 7 antioxidants-11-01137-f007:**
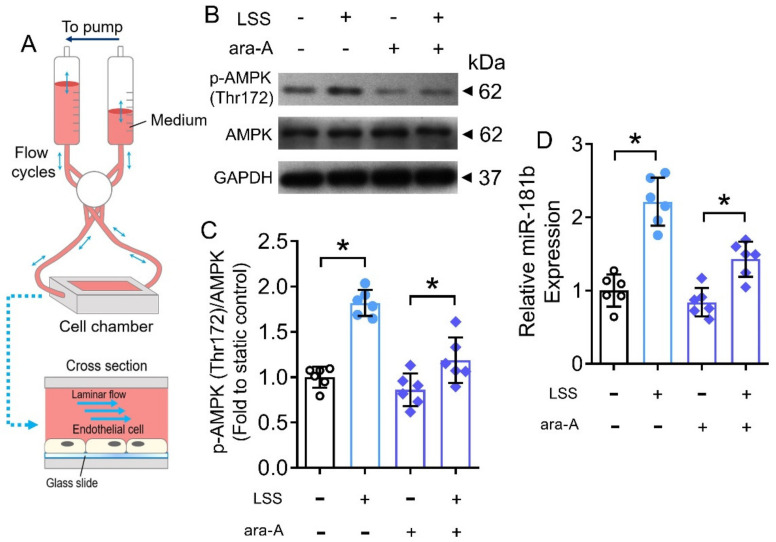
Effects of LSS on AMPK/miR-181b axis in endothelial cells. (**A**) Schematic diagram on the design of ibidi flow system and cross section of flow chamber. (**B**) Representative Western blots, and (**C**) quantification of Western blotting on expression of AMPK and p-AMPK at Thr172 in HUVECs exposed to LSS for 12 h. n = 6 per group. (**D**) RT-PCR on miR-181b expression in LSS-exposed HUVECs. n = 6 per group. Data are mean ± SD. * *p* < 0.05 (Brown-Forsythe and Welch ANOVA and Games-Howell’s multiple comparisons).

**Table 1 antioxidants-11-01137-t001:** Demographic characteristics of the included patients.

Variable	Mean ± SD
No. of patients	12
No. of diabetic/non-diabetic patients	4/8
No. of males/females	7/5
Mean age	63.75 ± 14.25

**Table 2 antioxidants-11-01137-t002:** List of human primer pairs used for quantitative RT-PCR.

mRNA/miRNA	Forward (5′-3′)	Reverse (5′-3′)	Accession Number
ICAM1	TTGGGCATAGAGACCCCGTT	GCACATTGCTCAGTTCATACACC	NM_000201
VCAM1	CAGTAAGGCAGGCTGTAAAAGA	TGGAGCTGGTAGACCCTCG	NM_001078
IL-6	CCTGAACCTTCCAAAGATGGC	TTCACCAGGCAAGTCTCCTCA	NM_000600
GAPDH	CCACTCCTCCACCTTTGAC	ACCCTGTTGCTGTAGCCA	NM_002046

**Table 3 antioxidants-11-01137-t003:** List of mouse primer pairs used for quantitative RT-PCR.

mRNA/miRNA	Forward (5′-3′)	Reverse (5′-3′)	Accession Number
Icam1	GTGATGCTCAGGTATCCATCCA	CACAGTTCTCAAAGCACAGCG	NM_010493
Vcam1	GTTCCAGCGAGGGTCTACC	AACTCTTGGCAAACATTAGGTGT	NM_011693
Il-6	TTCAGCCCTTGCTTGCCTC	ACACTTTTACTCCGAAGTCGGT	NM_031168
Cdh5	ATTGGCCTGTGTTTTCGCAC	CACAGTGGGGTCATCTGCAT	NM_009868
Gapdh	AGGTCGGTGTGAACGGATTTG	TGTAGACCATGTAGTTGAGGTCA	NM_001289726

## Data Availability

The data presented in this study are available in this manuscript.

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
