# Peer review of "Activation of AMPK/miR-181b Axis Alleviates Endothelial Dysfunction and Vascular Inflammation in Diabetic Mice"

_antioxidants, 2022, doi:10.3390/antiox11061137_

Round 1
Reviewer 1 Report
In this article, Cheng et al. investigate whether AMPK/miR-181b axis could be involved in diabetes-associated endothelial dysfunction. Using human clinical observation as a starting point (level of miR-181b in human diabetic renal arteries), the authors present in vitro and in vivo experiments to decipher the role of AMPK/miR-181b axis in endothelial dysfunction, vascular inflammation and oxidative stress in human endothelial cells and diabetic mice. Hey finally propose that chronic exercise, potentially through increased blood flow, activates AMPK/miR-181b axis in endothelial cells. All experiments are well document and clearly support the message.
Specific points:
- Page 3 and 4 – line 146 and 162: Did authors mean mmolL-1 instead of mmol-1 ?
- Figure 2 : Authors need to switch C and D panels
- Figure 6 and 7 (or in Mat/Meth part) authors should mentioned the number of WB used to quantify in Fig 6C and Fig 7C
- In the discussion part, authors should add a discussion on the downstream effectors of miR-181b potentially interesting in their model
Reviewer 2 Report
Dr. Cheng et al detected that miR-181b level decreased in the renal arteries of diabetic patients and suggested that miR-181b attenuated endothelial dysfunction through suppressing ROS, vascular inflammatory cytokine IL-6, and cell adhesion molecules I/VCAM-1s in arteries of diabetic mice. The authors suggested that AMPK activation up regulates miR-181b in HUVECs, and that chronic exercise may activate the AMPK to increase miR-181b in blood endothelial cells, possibly through increased blood shear flow. They used AMPK activator or inhibitors to address the possible link of AMPK and miR-181b in HUVEC. Most experiments have been designed and performed in suitable ways. There are several concerns in the present form of this manuscript.
Major concerns:
1. One important issue of the manuscript is the shear flow/AMPK/miR-181b axis in HUVECs (blood vessels). However, there are many other factors from the blood shear flow may affect AMPK activation, the shear flow might activate other signaling pathways which down regulate the miR-181b. For example, in Figure 6B or 7B, although the specific inhibitors were employed, have the authors ever tried the activation of other MAPK signals like ERK, PI3K or Akt phosphorylation, it would be very interesting if any of these are not affected.
2. What does it mean that the miR-18 activation decreases the adhesion molecules in blood endothelial cells? How these changes protect the diabetic mice?
Minor concerns:
1. Line 30: what’s the mean of “AICAR”? This reviewer would suggest replacing it with the “AMPK activator”.Line 81: “Endothelial cells…” is not accurate, “Blood endothelial cells…” would be better. There is no shear stress for lymphatic endothelial cells.
2. Line 81: “Endothelial cells…” is not accurate, “Blood endothelial cells…” would be better. There is no shear stress for lymphatic endothelial cells.
3. Figures 1, 4, 5: need to indicate normalizing housekeeping genes like HPRT, GAPDH etc.
4. Figure 2. Isn’t that “C” and “D” should be exchange? It looks like C is treated with AICAR; D is “transfection of miR-181b mimics”.
5. Lines 334-335 and in Figure 4, Icam1 and Vcam1 are not inflammatory markers, they are cell adhesion molecules.
6. Line 382, this reviewer is just curious about why here use the AMPK inhibitor compound C instead of another AMPK inhibitor Ara-A? Are there special reasons that have to use compound C in vivo?
7. Figure 7B the WB may need quantification.
Reviewer 3 Report
Comments to the author:
The communication submitted by Chak Kwong Cheng et al., studies whether miR-181b ameliorates diabetes-associated endothelial dysfunction in animal and in vitro models. The present work has been performed using a HUVEC cell line, mice (male), and human renal arteries. In general, the introduction, and figures are well presented. However, methods, results description, and conclusion must be improved to appreciate the relevance of these studies.
Major comments
Methods:
Humane arteries, line 97. The authors described that human renal arterial has been used. However, there is no description of age, gender, and whether or not glucose studies have been performed on humans. Since animal gender is male, the authors must add general information from human studies as I previously mentioned. Do the human arteries come from the same renal section?
Line 136. Could the Authors mention DMEM and FBS brand company and catalog number? In the same situation in line 185, the Authors must add the catalog number and brand of DMEM.
Cells culture section. Did the Authors test mycoplasma to cells previously to the studies?
Organ culture. There is no mention of how long the organ culture does it take? Line 171.
There is no immunostaining description in the methods sections. The authors must add this information.
In general, make sure to add the mining of all abbreviations, for example, EGM meaning, line 213.
RIPA, line 237. Add the RIPA buffer composition in a western blot section or company brand and catalog number. The Authors must add the catalog number of all antibodies used in the western blot section (lines 245-247).
Results:
In general, the authors might add columns number in the description of the results, for example, (Figure 1A, column two) to make sure that the readers follow the description.
Line 287, the Authors should add the observation time, ...was observed for 120 min (figure S1).
Taking into account chronic exercise experiments in mice, the author must add the equivalent time of exercise in humans in the discussion section.
The discussion section should be improved. For example, the authors can compare previous results in the bibliography, signaling pathways, and the relevance of miR-181b against the other member of this family. Why do they use renal arteries, Are there previous works using endothelial from other tissue? How specific is miR-181b to endothelial? Is there any previous work that reviews how to deliver miR-181B to the human or animal body, how specific is to the target, and does it affect other tissues?
